# Evaluating and Designing Sparse Autoencoders by Approximating Quasi-Orthogonality

**Sewoong Lee, Adam Davies**[*]**, Marc E. Canby**[*] **& Julia Hockenmaier**
Siebel School of Computing and Data Science
University of Illinois Urbana-Champaign
Illinois, IL 61801, USA
{samuel27, adavies4, marcec2, juliahmr}@illinois.edu

## Abstract

Sparse autoencoders (SAEs) are widely used in mechanistic interpretability research for large language models; however, the state-of-the-art method of using $k$-sparse autoencoders lacks a theoretical grounding for selecting the hyperparameter $k$ that represents the number of nonzero activations, often denoted by $\ell_0$. In this paper, we reveal a theoretical link that the $\ell_2$-norm of the sparse feature vector can be approximated with the $\ell_2$-norm of the dense vector with a closed-form error, which allows sparse autoencoders to be trained without the need to manually determine $\ell_0$. Specifically, we validate two applications of our theoretical findings. First, we introduce a new methodology that can assess the feature activations of pre-trained SAEs by computing the theoretically expected value from the input embedding, which has been overlooked by existing SAE evaluation methods and loss functions. Second, we introduce a novel activation function, top-AFA, which builds upon our formulation of approximate feature activation (AFA). This function enables top-$k$ style activation without requiring a constant hyperparameter $k$ to be tuned, dynamically determining the number of activated features for each input. By training SAEs on three intermediate layers to reconstruct GPT2 hidden embeddings for over 80 million tokens from the OpenWebText dataset, we demonstrate the empirical merits of this approach and compare it with current state-of-the-art $k$-sparse autoencoders. Our code is available at: https://github.com/SewoongLee/top-afa-sae.

## 1 Introduction

Language models pack meaning into dense vectors, but what if we could unpack them into separate, understandable pieces? To make this possible, sparse autoencoders (SAEs) have brought significant advances to mechanistic interpretability by demonstrating that dense embeddings inside language models can be effectively decomposed into a linear combination of human-interpretable feature vectors (Elhage et al., 2022; Bricken et al., 2023; Huben et al., 2023; Lieberum et al., 2024). Despite recent advances in sparse autoencoder evaluation, a crucial aspect has remained overlooked: the relationship between input embeddings and sparse feature vectors. Existing approaches enforce sparsity – either by limiting the number of active units or by penalizing the number of nonzero values – with little justification. In other words, although existing methods aim to construct sparse features corresponding to inputs, the selection of sparsity levels in these methods is independent of the inputs themselves. This missing link – between the input and its feature representation – is at the heart of what sparse autoencoders are supposed to recover and is the focus of this paper.

To address this issue, we take a fundamentally different approach: rather than evaluating SAEs based solely on sparsity *or* reconstruction error, we focus on the underlying *relationship* between input embeddings and their corresponding feature activations. This shift in per-

---

[*]These authors contributed equally to this work.

spective leads us to develop a new theoretical approximation for feature activation, along with practical tools for evaluating and designing sparse autoencoders under this framework. Our approach applies to pretrained SAEs commonly used in mechanistic interpretability, including models based on different sparsity penalties, such as GPT-2 Small (Bloom, 2024) and Gemma Scope (Lieberum et al., 2024). Furthermore, when training SAEs from scratch, our approach achieves reconstruction loss better than that of state-of-the-art $k$-sparse SAEs (Makhzani & Frey, 2013; Gao et al., 2024; Bussmann et al., 2024), without requiring the hyperparameter $k$ to be tuned. Our findings not only bridge a theoretical gap in current SAE evaluations but also open the door to novel avenues for experimental exploration.

The contributions of this paper are:

- We introduce Approximate Feature Activation (AFA), a closed-form estimation of the magnitude of sparse feature activations with provable error bounds.
- We present the ZF Plot to visualize and diagnose over- or under-activation of features based on the theoretical framework of AFA.
- We formalize $\varepsilon$-quasi-orthogonality as a geometric constraint arising from the superposition hypothesis, connecting it to the Johnson–Lindenstrauss Lemma, and propose $\varepsilon_{\mathrm{LBO}}$, the lower bound of quasi-orthogonality, a novel metric for evaluating SAE feature space.
- We propose top-AFA, a norm-matching activation function that adaptively selects the number of active features for each input vector without tuning $k$. Combined with a norm-matching loss $\mathcal{L}_{\mathrm{AFA}}$, this leads to a new SAE architecture, top-AFA SAE, that achieves better reconstruction performance compared to state-of-the-art top-$k$ and batch top-$k$ SAEs, while also offering stronger theoretical justification.

## 2 Preliminaries

**Notation**    We write vectors in bold as $\mathbf{v} = (v_1, v_2, \ldots, v_n)$ and refer to their scalar components $v_i$ in italics. We use $\mathbf{v}_i$ to denote the $i$-th vector in a collection of vectors, and use $\mathbf{v}(\cdot)$ to denote a function that returns a vector-valued output. $A_{i,j}$ denotes the $(i, j)$-th entry of a matrix $A$.

### 2.1 Sparse Autoencoders

The hidden embeddings of language models are densely packed with information, encoding overlapping features in high-dimensional space (Elhage et al., 2022). Individual dimensions do not correspond to specific, human-understandable concepts. To unpack these embeddings into more interpretable pieces, sparse autoencoders (SAEs) learn to decompose them into sparse activation vectors that weight learned feature directions.

Formally, we denote an input sequence by $\mathbf{x}$, which is transformed to an embedding vector $\mathbf{z}^{(l)}(\mathbf{x}) \in \mathbb{R}^d$ in LLMs, which is taken from the residual stream, the hidden embedding vector passed through the residual connections between transformer layers, of the last token index at layer $l$ (see Appendix B for details). This embedding is used as the input to the SAE, and we write it as $\mathbf{z}(\mathbf{x})$ or $\mathbf{z}$ for simplicity when the layer index and input are clear from context.

An SAE consists of an encoder and decoder, defined as:

$$\mathbf{f}(\mathbf{z}) = \sigma(W_{\mathrm{enc}}\mathbf{z} + \mathbf{b}_{\mathrm{enc}}), \quad \hat{\mathbf{z}} = W_{\mathrm{dec}}\mathbf{f}(\mathbf{z}) + \mathbf{b}_{\mathrm{dec}},$$

where $\mathbf{f}(\mathbf{z}) \in \mathbb{R}^h$ is a sparse latent vector and $\hat{\mathbf{z}} \in \mathbb{R}^d$ is the reconstruction of $\mathbf{z}$. The encoder matrix $W_{\mathrm{enc}} \in \mathbb{R}^{h \times d}$, decoder matrix $W_{\mathrm{dec}} \in \mathbb{R}^{d \times h}$, and biases $\mathbf{b}_{\mathrm{enc}} \in \mathbb{R}^h$, $\mathbf{b}_{\mathrm{dec}} \in \mathbb{R}^d$ are learned. The decoder matrix $W_{\mathrm{dec}}$ is often referred to as a *dictionary*, as it contains the directions for reconstructing the input from the feature space. The activation function $\sigma(\cdot)$ (e.g., ReLU or Top-$k$ (Makhzani & Frey, 2013)) enforces non-negativity or sparsity. Training minimizes a loss function that balances reconstruction and sparsity:

$$\mathcal{L}(\mathbf{z}) = \|\mathbf{z} - \hat{\mathbf{z}}\|_2^2 + \lambda_{\mathrm{sparsity}} S(\mathbf{f}(\mathbf{z})) + \alpha \mathcal{L}_{\mathrm{aux}}, \tag{1}$$

where $S(\mathbf{f})$ is a sparsity penalty (e.g., $\|\mathbf{f}\|_1$, $\|\mathbf{f}\|_0$).[1] $\lambda_{\text{sparsity}}$ controls the strength of the sparsity constraint, and $\mathcal{L}_{\text{aux}}$ is an auxiliary loss that prevents latent units from becoming inactive via Ghost Grads (Adam Jermyn, 2024).

## 2.2 Hypotheses Behind Sparse Autoencoders

$$\underbrace{\hat{\mathbf{z}} = W_{\text{dec}}\mathbf{f}(\mathbf{z}) + \mathbf{b}_{\text{dec}}}_{\text{Linear Representation}}, \quad \text{where} \quad \mathbf{z} \in \mathbb{R}^d,\ \mathbf{f}(\mathbf{z}) \in \mathbb{R}^h,\ \text{and} \quad \underbrace{h > d}_{\text{Superposition}} \tag{2}$$

As summarized in Equation (2), the design of sparse autoencoders is built on certain assumptions: the linear representation hypothesis and the superposition hypothesis. In this section, we formally present these two hypotheses and provide intuition behind their roles in motivating the architecture.

**Linear Representation.** In this paper, unless otherwise specified,[2] we use the term *linear representation hypothesis (LRH)* with the following definition as discussed in Elhage et al. (2022); Bricken et al. (2023); Nanda et al. (2023); Lieberum et al. (2024); Smith (2024); Engels et al. (2024b;a); Laptev et al. (2025):

> **Linear Representation Hypothesis** Internal activations of neural networks can be represented as linear combinations of vectors. Formally, for a given input $\mathbf{x}$ to the neural network model, hidden embeddings $\mathbf{z}(\mathbf{x}) \in \mathbb{R}^d$ can be represented by a linear transformation[a] of a feature activation vector $\mathbf{f} \in \mathbb{R}^h$ with a constant weight matrix $W \in \mathbb{R}^{d \times h}$,
>
> $$\mathbf{z}(\mathbf{x}) = W\mathbf{f}$$
>
> ---
> [a]This hypothesis also holds under affine transformations if we define $\mathbf{f}' = [\mathbf{f}^\top \mid \|\mathbf{b}\|_2]^\top$, where $\mathbf{b}$ is the translation bias.

**Superposition.** In machine learning interpretability studies, the term *superposition* refers to the phenomenon in which models represent more features than the dimensionality of their representations (Elhage et al., 2022). The statement that there can be more features than neurons in neural networks implies that neural network models can exploit the property of high dimension supported by the Johnson-Lindenstrauss lemma (Lindenstrauss & Johnson, 1984). To formally state:

> **Superposition Hypothesis.** Let $\mathbf{z}(\mathbf{x}) \in \mathbb{R}^d$ denote the hidden embeddings for a given input $\mathbf{x}$ to an LLM. There exists a function $\Phi : \mathbb{R}^h \to \mathbb{R}^d$ for feature vectors $\mathbf{f} \in \mathbb{R}^h$ such that:
>
> $$\mathbf{z}(\mathbf{x}) = \Phi(\mathbf{f}), \text{ where } h > d.$$

## 2.3 Theoretical Tools for High-Dimensional Analysis

**Theorem 1** (Johnson-Lindenstrauss Lemma). *Let $Q \subset \mathbb{R}^h$ be a set of $n$ points. For $\delta \in (0, 1/2)$ and $d = \frac{20 \ln n}{\delta^2}$, there exists a linear mapping $\Psi : \mathbb{R}^h \to \mathbb{R}^d$ such that $\forall \mathbf{u}, \mathbf{v} \in Q$:*

$$(1 - \delta)\|\mathbf{u} - \mathbf{v}\|_2^2 \leq \|\Psi(\mathbf{u}) - \Psi(\mathbf{v})\|_2^2 \leq (1 + \delta)\|\mathbf{u} - \mathbf{v}\|_2^2.$$

*Proof of Theorem.* See Lindenstrauss & Johnson (1984); Vempala (2005); Kakade (2010). □

---

[1]The $\ell_0$-norm of a vector $\mathbf{f}$, denoted as $\|\mathbf{f}\|_0$, counts the number of non-zero elements in $\mathbf{f}$. Although it is not a norm in the mathematical sense, it is widely referred to as $\ell_0$ "norm" in this context due to its usefulness in expressing sparsity.

[2]For more details on the LRH, see Appendix C.

**Definition 1** (*ε-Quasi-Orthogonal Set* (Kainen & Kůrková, 1993; Kainen & Kůrková, 2020)).
*Let $S^{d-1} = \{\mathbf{v} \in \mathbb{R}^d : \|\mathbf{v}\|_2 = 1\}$ be the unit sphere in the d-dimensional Euclidean space. For $\varepsilon \in [0, 1)$, a subset $T \subset S^{d-1}$ is called an ε-**quasi-orthogonal set** if:*

$$\mathbf{u} \neq \mathbf{v} \in T \Rightarrow |\mathbf{u} \cdot \mathbf{v}| \leq \varepsilon.$$

## 3   Related Work

**Finding Interpretable Dictionaries.**   Initially, Lee Sharkey (2022) investigated a toy setting where dense embeddings were explicitly constructed from combinations of ground-truth sparse feature vectors. This reversed setup enabled testing whether SAEs can recover known ground-truth features. The key finding was that, given a properly tuned $\ell_1$ sparsity coefficient, SAEs could accurately recover the ground-truth components, identifying a "Goldilocks zone" for sparsity regularization. This foundational insight later inspired follow-up work by Huben et al. (2023), extending the idea to real-world settings, showing that SAEs trained on models like Pythia-70M can discover highly interpretable features.

**Pre-trained SAEs: GPT-2 Small and Gemma Scope.**   Building on these insights, Bloom (2024) successfully extended sparse autoencoder training to GPT-2 Small (Radford et al., 2019), releasing a pretrained SAE model widely used in follow-up research. While retaining the $\ell_1$ sparsity penalty, their model introduced several key architectural improvements: ghost gradients for addressing inactive features and a unit-norm decoder. GPT-2 Small has since become a standard model for SAE research due to its manageable size and representative embedding structure, and has been studied in follow-up works such as Chaudhary & Geiger (2024); Bussmann et al. (2024); Minegishi et al. (2025).

Lieberum et al. (2024) introduced the Gemma Scope SAE, trained on Gemma 2's hidden embeddings (Gemma Team, 2024). Unlike prior SAEs that relied on $\ell_1$ sparsity, Gemma Scope adopted an $\ell_0$ penalty to reduce *shrinkage effects* (Rajamanoharan et al., 2024), which had previously led to suppression of overall activations. Furthermore, they employed JumpReLU, an activation function that applies a non-zero threshold to mitigate interference caused by superposition. The resulting model has become one of the most widely adopted SAEs in the interpretability community (Lin, 2023).

**Top-$k$ Activation.**   Gao et al. (2024) demonstrated the effectiveness of $k$-sparse autoencoders (Makhzani & Frey, 2013), which use a sparsity-enforcing activation function that retains only the $k$ largest pre-activations per input. This approach eliminates the need for a separate sparsity penalty so that $\lambda_{\text{sparsity}} = 0$ in Equation (1). They also discovered a *scaling law* between sparsity, measured by $\ell_0$, and reconstruction error, measured by mean square error (MSE), suggesting that there exists a empirical boundary that SAEs cannot surpass. However, the main limitation of top-$k$ is its reliance on a fixed $k$, which lacks a principled justification. To address this, Bussmann et al. (2024) proposed batch top-$k$, a variant that enforces sparsity on average across a batch rather than per input. Their method enables dynamic per-input activation while maintaining a fixed average sparsity, resulting in better reconstruction.

**Evaluation of Sparse Autoencoders**   The question of what makes a good SAE has been actively explored and remains open. Till (2024) explained that since there are infinitely many ways to decompose a dense embedding, failing to ensure near-orthogonality may hinder the recovery of the true underlying features. Although this work proposed orthogonality as a key geometric criterion, it lacked formalization or empirical validation. Simple structural metrics, such as pairwise cosine similarity between decoder vectors, are easy to compute but remain model-centric and fail to capture data-driven behavior. To address this, recent efforts, such as SAE Bench (Karvonen et al., 2025), have introduced both supervised (e.g., using LLM judges) and unsupervised metrics to evaluate SAE quality from an input-driven perspective.

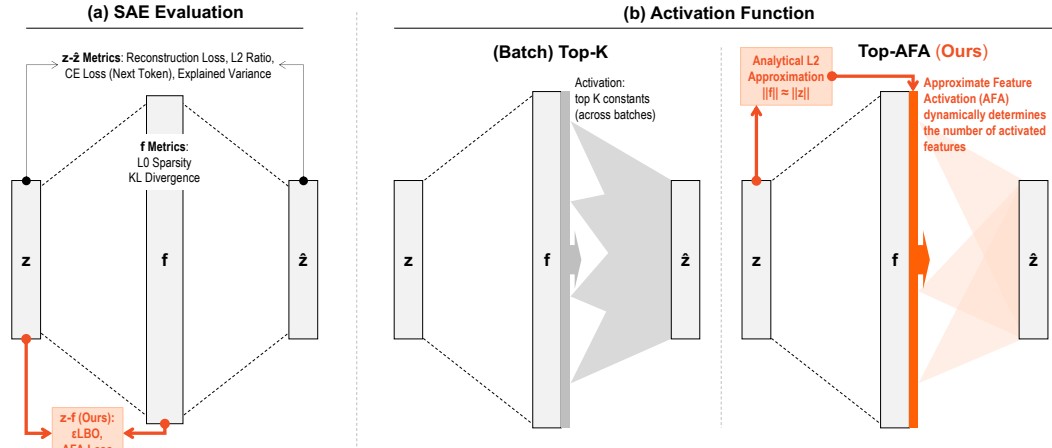

Figure 1: **(a) Comparison of SAE evaluation approaches.** Existing unsupervised metrics, such as SAE Bench (Karvonen et al., 2025), assess either reconstruction quality (i.e., the relationship between **z** and **ẑ**) or the sparsity of **f** in isolation. Our proposed AFA-based evaluation fills this gap by introducing a way to assess the alignment between input embeddings and feature activations. **(b) Comparison of fixed-$k$ activation (e.g., top-$k$, batch top-$k$) with our adaptive top-AFA.** While top-$k$ uses a constant sparsity level, top-AFA dynamically selects the number of active features per input by matching the activation norm with the input norm.

## 4 Approximating Feature Activation

Can we define an input-driven, theoretically justifiable metric that reflects feature activation quality, even under superposition? In this section, we define Approximate Feature Activation (AFA) as the closed-form solution that approximates the $\ell_2$ norm of a ground-truth sparse feature vector. In our study, we derive a specific form called *linear-AFA* under the simplified setting where the linear representation hypothesis is assumed to hold – e.g., in SAEs with a one-layer decoder, as commonly used in recent work (Bloom, 2024; Lieberum et al., 2024; He et al., 2024; Nabeshima, 2024; Bart Bussmann, 2024; Gao et al., 2024; Bussmann et al., 2024).

First, we propose the notion of an $\varepsilon$-quasi-identity matrix, denoted as $I^{\leq\varepsilon}$, which utilizes the definition of $\varepsilon$-quasi-orthogonality and serves as a useful tool for quantifying superposition in high-dimensional spaces.

**Definition 2** ($\varepsilon$-Quasi-Identity Matrix). *A matrix $I^{\leq\varepsilon} \in \mathbb{R}^{h\times h}$ is called an $\varepsilon$-**quasi-identity matrix** if it satisfies the following properties:*

$$I_{i,j}^{\leq\varepsilon} = \begin{cases} 1, & \text{if } i = j, \\ \varepsilon_{i,j}, & \text{if } i \neq j, \text{ where } |\varepsilon_{i,j}| \leq \varepsilon, \end{cases}$$

*where each $\varepsilon_{i,j}$ is a scalar satisfying $|\varepsilon_{i,j}| \leq \varepsilon$, for some fixed $\varepsilon \in [0, 1)$ which represents the maximum off-diagonal deviation from the identity matrix for all $i, j \in [h]$.*

Now, let $D \in \mathbb{R}^{d\times h}$ be a decoder (dictionary) matrix. If its columns are normalized, then the Gram matrix $D^\top D$ becomes the $\varepsilon$-quasi-identity matrix $I^{\leq\varepsilon}$ for some $\varepsilon$, which is the maximum inner product magnitude between distinct dictionary column vectors. Formally,

**Definition 3** (Quasi-Orthogonality of a Dictionary). *We define the **quasi-orthogonality** of a dictionary $D$ as:*

$$\varepsilon := \max_{i\neq j} \left| \frac{D_{\cdot,i}^\top D_{\cdot,j}}{\|D_{\cdot,i}\|_2 \cdot \|D_{\cdot,j}\|_2} \right|.$$

In practice, pre-trained SAE decoders often do not satisfy this unit-norm constraint. However, without loss of generality, we can construct an equivalent decomposition by normalizing each decoder column and scaling the index of the corresponding feature vector with the decoder column's norm. This transformation allows us to formalize the following theorem:

**Theorem 2** (Linear-AFA). *For a given input* $\mathbf{x}$, *if an embedding vector* $\mathbf{z}(\mathbf{x}) \in \mathbb{R}^d$ *satisfies the linear representation hypothesis (LRH), then* $\|\mathbf{z}(\mathbf{x})\|_2^2$ *approximates* $\|\mathbf{f}\|_2^2$, *the square of* $\ell_2$-*norm of feature activations* $\mathbf{f} \in \mathbb{R}^h$, *with an error bound* $\|\mathbf{f}\|_2^2 \in \left[ \frac{\|\mathbf{z}(\mathbf{x})\|_2^2}{1+\varepsilon(h-1)}, \frac{\|\mathbf{z}(\mathbf{x})\|_2^2}{1-\varepsilon(h-1)} \right]$, *where* $\varepsilon$ *is the quasi-orthogonality of the dictionary* $D \in \mathbb{R}^{d \times h}$ *used by the features, such that* $\mathbf{z}(\mathbf{x}) = D \cdot \mathbf{f}$.

*Proof of Theorem.* Since $D$ is a $\varepsilon$-quasi-orthogonal dictionary, $D^\top D = I^{\leq \varepsilon}$, where $I^{\leq \varepsilon}$ is a $\varepsilon$-quasi-identity matrix. Then,

$$\|\mathbf{z}(\mathbf{x})\|_2^2 = \mathbf{z}(\mathbf{x})^\top \mathbf{z}(\mathbf{x}) = (D\mathbf{f})^\top (D\mathbf{f}) = \mathbf{f}^\top (D^\top D)\mathbf{f} = \mathbf{f}^\top (I^{\leq \varepsilon})\mathbf{f} = \sum_{i=1}^h \sum_{j=1}^h f_i f_j I_{i,j}^{\leq \varepsilon}.$$

By using Definition 2 and applying Cauchy-Schwarz (3),

$$\left| \|\mathbf{z}(\mathbf{x})\|_2^2 - \|\mathbf{f}\|_2^2 \right| = \left| \sum_{i \in [h]} \sum_{j \in [h]} f_i f_j I_{i,j}^{\leq \varepsilon} - \sum_{i \in [h]} f_i^2 \right| \leq \sum_{i \in [h]} \sum_{\substack{j \in [h]; \\ j \neq i}} |f_i||f_j||\varepsilon_{i,j}| \leq \varepsilon(\|\mathbf{f}\|_1^2 - \|\mathbf{f}\|_2^2)$$

$$\leq \varepsilon(h\|\mathbf{f}\|_2^2 - \|\mathbf{f}\|_2^2) = \varepsilon(h-1)\|\mathbf{f}\|_2^2 \Rightarrow \|\mathbf{z}(\mathbf{x})\|_2^2 \in \left[ (1 - \varepsilon(h-1))\|\mathbf{f}\|_2^2, \ (1 + \varepsilon(h-1))\|\mathbf{f}\|_2^2 \right].$$

$$\therefore \|\mathbf{f}\|_2^2 \in \left[ \|\mathbf{z}(\mathbf{x})\|_2^2 / (1 + \varepsilon(h-1)), \ \|\mathbf{z}(\mathbf{x})\|_2^2 / (1 - \varepsilon(h-1)) \right].$$

$\square$

## 5  How to Measure $\varepsilon$: Quasi-Orthogonality of Dictionary

Although Theorem 2 provides a principled connection between $\mathbf{z}$ and $\mathbf{f}$, it relies on the quasi-orthogonality constant $\varepsilon$, which is unknown in practice. To address this, we introduce two approaches to obtain the range of $\varepsilon$.

**Upper Bound Approach based on the JL Lemma.**  The JL lemma (Theorem 1) has often been suggested as a mathematical intuition that machine learning models gain an advantage in high-dimensional representation due to the exponential growth of almost orthogonal vectors (Elhage et al., 2022; Ghilardi et al., 2024). However, there is a lack of formulation regarding the closed form bound using the JL lemma. By deriving $|\varepsilon| \leq \delta = \sqrt{20 \ln[h]/d}$ from the JL lemma using the method described in Appendix F, we can define:

$$\varepsilon_{\text{JL}} := \sqrt{\frac{20 \ln h}{d}}.$$

Although the JL lemma ensures the existence of a quasi-orthogonal basis within a bounded error, we find that pretrained SAE decoders often exceed this bound (see Appendix G). This reveals a key limitation: even state-of-the-art SAEs fail to learn decoders that achieve the theoretically permitted level of quasi-orthogonality, making their use unreliable.

**Lower Bound Approach with Pre-trained SAE Features.**  A tighter and more practical bound can be obtained from SAE feature vectors for each input, from the lower end. The closed-form error bound derived in Theorem 2 provides a way to estimate the quasi-orthogonality constant $\varepsilon$. Specifically, we use a lower bound of $\varepsilon$, based on the inequality $\left| \|\mathbf{z}(\mathbf{x})\|_2^2 - \|\mathbf{f}\|_2^2 \right| / (h-1)\|\mathbf{f}\|_2^2 \leq \varepsilon$. However, this time, the ground-truth $\mathbf{f}$ is not accessible.

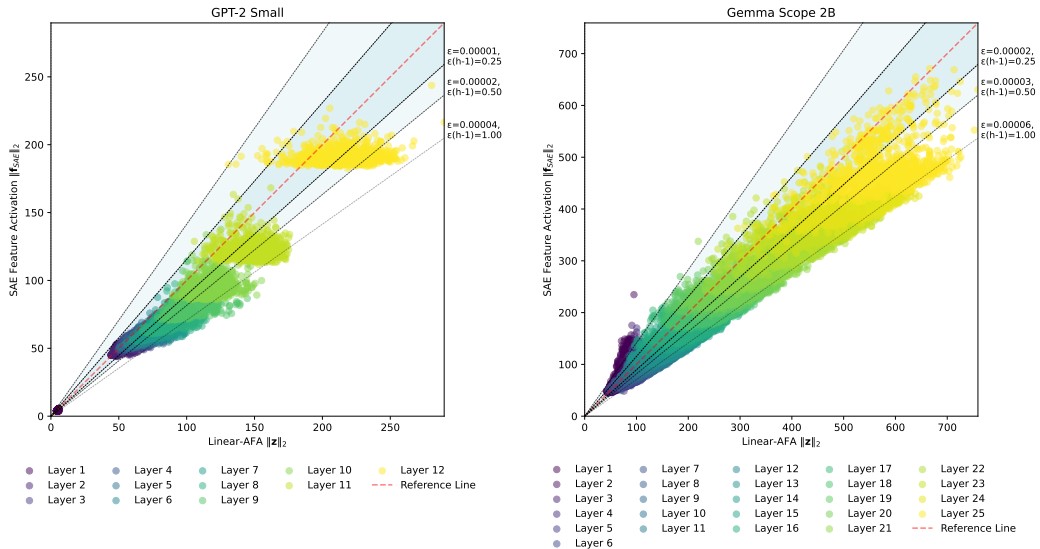

Figure 2: **ZF Plots.** ZF plots shows the relationship between dense embedding vector norm ($\|\mathbf{z}\|_2$) and the learned feature activation norm ($\|\mathbf{f}\|_2$) by layers. Each point corresponds to a sequence input to the language model, allowing us to compare the magnitude and error direction (i.e., over- or under-activation) of the SAE's features compared to the red dashed reference line, input by input (For more details, see Fig. 3). The shaded region visualizes the error bound based on $\varepsilon(h-1)$ derived in Theorem 2, where $\varepsilon$ is the quasi-orthogonality of dictionaries and $h$ is the dimensionality of the corresponding SAE feature vector. These plots are drawn using 1k input sequences of length 128 from the OpenWebText dataset. The phenomenon of increasing norm in higher layers is explained in Hex & Turn (2023).

In this case, we can leverage the fact that SAEs are designed to estimate $\mathbf{f}$, and denote this estimate as $\mathbf{f}_{\mathrm{SAE}}$. Thus, we can compute the epsilon lower bound:

$$\varepsilon_{\mathrm{LBO}}(\mathbf{x}) := \frac{\left| \|\mathbf{z}(\mathbf{x})\|_2^2 - \|\mathbf{f}_{\mathrm{SAE}}(\mathbf{x})\|_2^2 \right|}{(h-1)\|\mathbf{f}_{\mathrm{SAE}}(\mathbf{x})\|_2^2}$$

In layman's terms, this bound represents the minimum $\varepsilon$ of quasi-orthogonality that any decoder must have in order to reconstruct the observed feature activations from the embedding. Since $\varepsilon_{\mathrm{LBO}}$ is computed directly from the feature vectors – the very structure that SAEs ultimately aim to uncover – it offers a decoder-agnostic alternative that is free from the decoder-induced noise discussed in the upper bound approach.

## 6 Evaluation Metric: Missing Link between z and f

Despite recent advances in SAE evaluation methods such as SAE Bench (Karvonen et al., 2025), Figure 1-(a) shows a key limitation in current metrics: they either evaluate the relationship between the input embedding $\mathbf{z}$ and reconstructed embedding $\hat{\mathbf{z}}$, or they measure the sparsity of the feature vector $\mathbf{f}$ alone. Attempts to bridge $\mathbf{z}$ and $\mathbf{f}$ have remained limited to measuring decoder cosine similarities, which do not assess their actual relationship based on inputs. In contrast, our approach introduces an evaluation metric that directly links $\mathbf{z}$ and $\mathbf{f}$, enabling us to assess whether the feature activations are appropriately aligned with the input representation. This relationship between $\mathbf{z}$ and $\mathbf{f}$ can be visualized using pretrained SAEs, as shown in Figure 2. The geometrical intuition that is explained in Figure 3 allows for a fundamentally different perspective: instead of treating sparsity as an isolated property, we evaluate whether the activations themselves are justified by the input.

If we apply this finding to the evaluation of GPT-2 Small and Gemma Scope 2B, Figure 4 shows that $\varepsilon$LBO offers a different perspective compared to mean squared error (MSE),

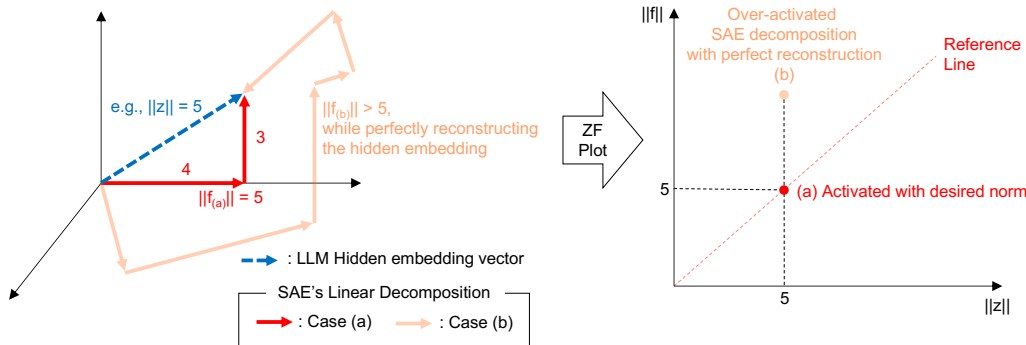

Figure 3: **Geometrical intuition behind ZF plots.** A dense embedding vector (blue dashed) can be perfectly reconstructed using an over-activated set of SAE features (light orange solid), even when their norm is misaligned. This leads to a discrepancy between the norm of the input ($\|\mathbf{z}\|$) and the reconstructed activation norm ($\|\mathbf{f}\|$). On the right, the ZF plot visualizes this mismatch per input, with the red dashed line indicating ideal alignment. Case (b) shows an over-activated decomposition; Farther distance from the reference line indicates either excessive activation despite seemingly good reconstruction or the absence of a low-$\varepsilon$ dictionary that can reconstruct the input.

---

**Algorithm 1** Top-AFA Activation Function

---

**Require:** $\mathbf{z} \in \mathbb{R}^{B \times d}$, $W_{\text{enc}} \in \mathbb{R}^{d \times h}$, $W_{\text{dec}} \in \mathbb{R}^{h \times d}$, $\mathbf{b}_{\text{enc}} \in \mathbb{R}^d$, $\mathbf{b}_{\text{dec}} \in \mathbb{R}^d$       $\triangleright$ $B$: batch size
**Ensure:** Reconstructed embedding $\hat{\mathbf{z}}$
 1: $\mathbf{z}_{\text{cent}} \leftarrow \mathbf{z} - \mathbf{b}_{\text{enc}}$
 2: $a \leftarrow \|\mathbf{z}_{\text{cent}}\|_2^2$
 3: $\mathbf{f} \leftarrow \text{ReLU}(\mathbf{z}_{\text{cent}} W_{\text{enc}})$
 4: $\mathbf{s} \leftarrow (\mathbf{f} \odot \|W_{\text{dec}}\|_2)^2$                          $\triangleright$ $\odot$: elementwise multiplication
 5: $\pi \leftarrow \text{argsort}(\mathbf{s}, \text{descending})$
 6: $C \leftarrow \text{cumsum}(\mathbf{s}[\pi])$; set $C[-1] \leftarrow \kappa$                    $\triangleright$ $\kappa$ is a large constant
 7: $a_{\text{target}} \leftarrow \sqrt{a}$, $C \leftarrow \sqrt{C}$
 8: $k \leftarrow \text{argmin}_k\{|C_k - a_{\text{target}}|\} + 1$
 9: Construct mask $\mathbf{m}$ that retains the top $k$ indices in $\pi$
10: $\mathbf{f}_{\text{topk}} \leftarrow \mathbf{f} \odot \mathbf{m}$
11: $\hat{\mathbf{z}} \leftarrow \mathbf{f}_{\text{topk}} W_{\text{dec}} + \mathbf{b}_{\text{dec}}$
12: **return** $\hat{\mathbf{z}}$

---

which is the most commonly used metric for evaluating sparse autoencoders (Gao et al., 2024; Bussmann et al., 2024). While overall trends are similar, there are differences in distributional shape (e.g., long tails), modality (e.g., unimodal vs. bimodal), and ranking order (MSE-based ranking vs. $\varepsilon$LBO-based ranking).

## 7 Designing a Novel Sparse Autoencoder

**Activation Function.** Top-$k$ based activations suffer from a fundamental limitation: the expected sparsity level $E[\ell_0]$ remains unknown and theoretically unjustified (Gao et al., 2024). To address this, we propose the top-AFA, an activation function that adaptively selects the number of active features per input by matching the activation norm to a theoretically grounded target derived from the input embedding norm. This removes the need to manually set or estimate $k$, and instead provides an indirect estimate of $E[\ell_0]$. The core idea behind top-AFA is to activate the minimum number of features such that the sparse feature vector norm approximates the AFA target $\|\mathbf{z}\|_2$, as described in Figure 1-(b) and Algorithm 1. From this perspective, sparsity and reconstruction are no longer in a trade-off relationship. Instead, sparsity can be chosen *for* reconstruction.

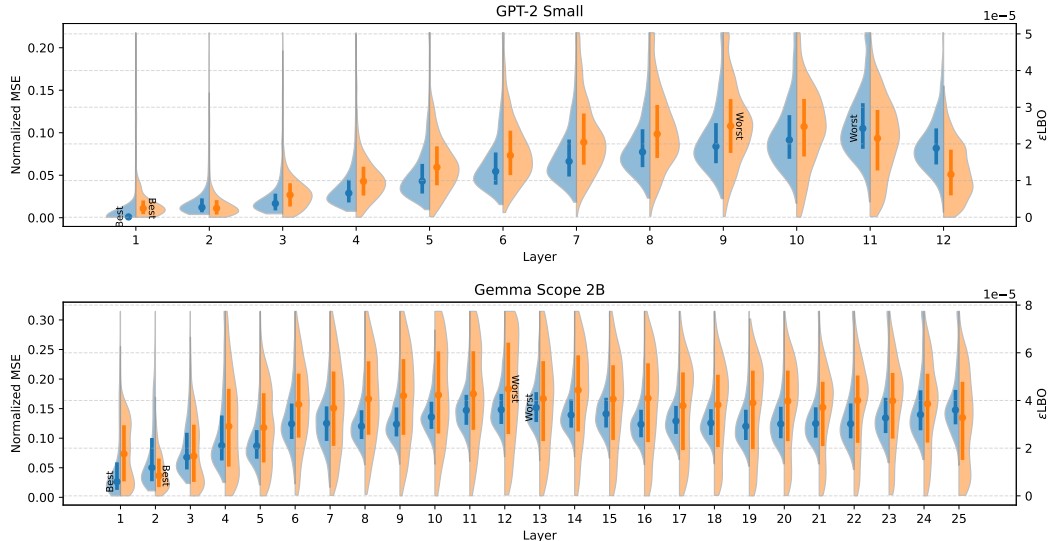

Figure 4: **Violin Plots for Comparing Different Evaluations of SAEs** (Lower values are better). Blue violins (left) represent the normalized MSE (NMSE), which reflects reconstruction loss, while orange violins (right) correspond to $\varepsilon$LBO, which measures quasi-orthogonality. Distributions are computed over 1k input sequences of length 128 from the OpenWeb-Text dataset, and the region of each violin captures 99% of the distribution to mitigate the influence of extreme outliers. Higher $\varepsilon$LBO values imply that no sufficiently orthogonal dictionary can explain the feature activation, suggesting poor separation among the activated features. While overall trends across layers are similar, the best- and worst-ranked layers and the distributional shapes vary, showing differences between NMSE and $\varepsilon$LBO. These results show that the two evaluation metrics yield different evaluations of the same models.

**Loss Function.** Can we design an SAE loss function that accounts for the amount of feature activation expected based on each input vector? A natural option would be to directly use $\varepsilon$LBO; however, it has certain limitations when applied as a training loss (see Appendix H). Given that our key theoretical takeaway is $\lim_{\varepsilon \to 0} \|\mathbf{f}\|_2 = \|\mathbf{z}\|_2$, we define the following simple loss function: $\mathcal{L}_{\text{AFA}} = (\|\mathbf{f}\|_2 - \|\mathbf{z}\|_2)^2$. Adding $\mathcal{L}_{\text{AFA}}$ to the standard loss in Equation (1) yields the following objective:[3]

$$\mathcal{L}(\mathbf{x}) = \|\mathbf{x} - \hat{\mathbf{x}}\|_2^2 + \alpha\mathcal{L}_{\text{aux}} + \lambda_{\text{AFA}}\mathcal{L}_{\text{AFA}}.$$

Note that this additional loss term does not shift the optimum of the reconstruction objective, since the $\ell_2$ reconstruction loss is minimized when $D\mathbf{f} = \mathbf{z}$, and under this condition, Theorem 2 guarantees that $\mathbb{E}_\varepsilon[\|\mathbf{f}\|_2] = \|\mathbf{z}\|_2$.

**Experimental Results.** We follow the experimental setup of Bussmann et al. (2024), using $h = 16 \times d$, but extend the evaluation to layers 6 and 7 in addition to layer 8. Figure 5 presents results for GPT-2, comparing Top-AFA with top-$k$ and batch top-$k$. Notably, Top-AFA achieves better reconstruction performance than top-$k$-based baselines and exceeds the scaling law boundary observed by Gao et al. (2024).[4] This suggests that adaptively selecting activations based on AFA can lead to improvements beyond what fixed-sparsity approaches can achieve. However, as shown in Appendix I, the effectiveness of Top-AFA depends on proper tuning of the coefficient associated with the AFA loss $\mathcal{L}_{\text{AFA}}$, and a stable coefficient value of 1/16 was observed across all layers.[5]

---

[3]We follow prior work in setting the auxiliary loss weight $\alpha$ to 1/32, a commonly used value in SAE training (Gao et al., 2024; Bussmann et al., 2024).

[4]Layer 8 was used in Bussmann et al. (2024) to validate the performance of batch top-$k$.

[5]While Top-AFA introduces a new hyperparameter $\lambda_{\text{AFA}}$, it differs fundamentally from the fixed sparsity hyperparameter $k$ used in top-$k$ activations. Specifically, $\lambda_{\text{AFA}}$ is a training-time regularization

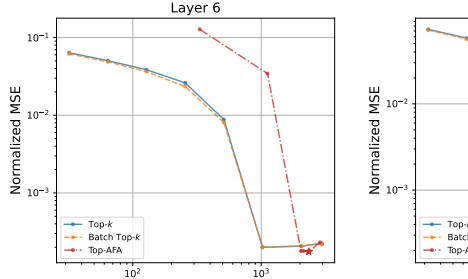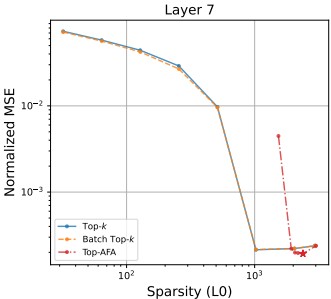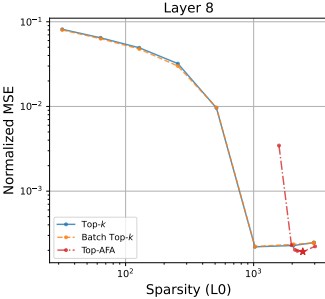

Figure 5: Reconstruction performance of Top-AFA compared to top-*k* and batch top-*k* activation functions on GPT-2 layers 6, 7, and 8. These layers were chosen to represent the model's middle depth, with layer 6 positioned at the center of GPT-2's 12-layer architecture. Layer 8 was also used in Bussmann et al. (2024), allowing for a direct comparison with prior work. The SAE with minimum reconstruction loss at each layer is marked with a star (⋆). Top-AFA outperforms other baselines on all tested layers, where its MSE (reconstruction loss) surpasses the scaling law boundary reported by Gao et al. (2024) These findings demonstrate that adaptively selecting activations can surpass the limitations of fixed-sparsity approaches. Detailed settings and results are provided in Appendix I.

## 8   Conclusion

We present a new theoretical and practical framework for SAEs by addressing a core limitation of top-*k* activations: the lack of principled *k* selection. Our Approximate Feature Activation (AFA) formulation provides a closed-form estimate of the $\ell_2$-norm of sparse activations, leading to $\varepsilon_{\mathrm{LBO}}$, the first metric linking input embeddings to activation magnitudes. Building on this, we propose top-AFA SAE, an SAE that adaptively selects active features by matching activation norms to dense embedding norms, removing the need to tune *k*. Experiments on GPT-2 show that top-AFA SAE outperforms top-*k* and batch top-*k* SAE. In conclusion, our research sheds light on a key limitation in the existing literature – the inaccessibility of $\mathbb{E}[\ell_0]$ – by taking a simple yet novel approach: approximating $\mathbb{E}[\ell_2]$ ultimately provides a path towards obtaining $\ell_0$.

## Acknowledgements

We thank Atticus Geiger for discussions with Adam Davies that influenced our early thinking on this work. This research was supported in part by award HR0011249XXX from the U.S. Defense Advanced Research Projects Agency Friction for Accountability in Conversational Transactions (FACT) program and the Illinois Computes project which is supported by the University of Illinois Urbana-Champaign and the University of Illinois System, and used the Delta advanced computing and data resource which is supported by the National Science Foundation (award OAC 2005572) and the State of Illinois. Delta is a joint effort of the University of Illinois Urbana-Champaign and its National Center for Supercomputing Applications. Adam Davies is supported by the NSF and the Institute of Education Sciences, U.S. Department of Education, through Award #2229612 (National AI Institute for Inclusive Intelligent Technologies for Education). Any opinions, findings, and conclusions or recommendations expressed in this material are those of the author(s) and do not necessarily reflect the views of DARPA, the NSF, or the U.S. Department of Education.

---

coefficient whose influence is limited to the optimization phase, whereas *k* directly relates to inference-time because it refers to the number of active features for each input example.

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

## A  Limitations and Future Work

Theoretically, our analysis is limited to the linear representation hypothesis; extending AFA to more general superposition settings – such as when $\Phi(\cdot)$ is a two-layer neural network or other recent SAE variants such as Matryoshka SAEs (Nabeshima, 2024; Bart Bussmann, 2024) – remains an open question. The JL lemma does not play a key role, but depending on the assumptions and claims, there are multiple versions of it that can be used (Har-Peled, 2005). It's also possible to explore tighter versions with more assumptions that we haven't used. Additionally, we do make a simplifying assumption by treating the embedding norm as its expectation; removing this assumption could lead to a more rigorous but possibly looser bound.

Empirically, in Figure 5, we identified two limitations in the design of top-AFA SAE: (1) the necessity of introducing a new hyperparameter $\lambda_{\text{AFA}}$ instead of $k$ and (2) the resulting high $\ell_0$ values in this setting. Despite these limitations, our approach allows the model to adaptively determine the effective number of active features for each input, rather than enforcing a fixed $k$ across all inputs. This reflects the intuition that different examples may require different amounts of feature activation, which a fixed $k$ cannot capture. Nevertheless, a deeper investigation into the necessity of the loss coefficient remains a direction for future work.

## B  Additional Terminology

**Residual Stream**  The residual stream refers to the hidden embedding vectors that flows through transformer layers via residual connections, typically updated as $\mathbf{z}^{(l+1)}(\mathbf{x}) = \mathcal{T}^{(l)}(\mathbf{z}^{(l)}(\mathbf{x})) + \mathbf{z}^{(l)}(\mathbf{x})$, where $\mathcal{T}^{(l)}(\cdot)$ is the $l$-th Transformer block (Vaswani et al., 2017) in language models.

**Overcompleteness**  Dictionary is sometimes referred to as an "overcomplete basis" due to the intuition of using more vectors than the dimension. However, since a basis cannot be overcomplete by definition, we adopt the more precise term "dictionary" in this paper.

## C  Further Classification of LRH

It is notable that the validity of the linear representation hypothesis (LRH) depends on factors such as the dictionary size $h$ relative to the embedding dimension $d$, and the domain of inputs $\mathbf{x}$ over which it is applied. For instance, in the limit as $h \to \infty$, each column of $W$ can memorize a specific input, and the feature vector $\mathbf{f}$ can trivially become one-hot vector (Gao et al., 2024; Smith, 2024). A non-trivial hypothesis is when the dictionary size $h$ is constrained. In practical implementation of SAEs, $h = 16 \times d$ in GPT-2 Small (Bloom, 2024), whereas in Gemma Scope 2B, it ranges from 8 to 28 times the embedding dimension (Lieberum et al., 2024).

Depending on the range of $\mathbf{x}$, we can classify LRH assumptions as follows:[6]

- **Weak LRH:** This version states that **some** features, though not necessarily all, can be represented as a linear combination human-interpretable features. Weak LRH has been widely supported by research (Nanda et al., 2023; Lieberum et al., 2024).

- **Strong LRH:** This version states that **all** features can be represented as a linear combination. However, this stronger claim has been widely disproved Smith (2024); Engels et al. (2024a).

**Comparison between SH and LRH.** In both cases, the feature vector $\mathbf{f} \in \mathbb{R}^h$ is assumed to result from a latent decomposition of the embedding vector $\mathbf{z} \in \mathbb{R}^d$, although the ground truth of such decompositions may not exist in the real-world language models. Under the SH, there are no structural assumptions on the encoder function $\Phi$, except that the feature space is higher-dimensional than the input space (i.e., $h > d$). The LRH, in contrast, assumes that $\Phi(\cdot)$ is a linear matrix $W \in \mathbb{R}^{d \times h}$ but does not place any constraints on the relative dimensionality of the feature and input spaces.

## D Proof Tools

**Theorem 3** ($\ell_1$-$\ell_2$ Norm Inequality)**.** *For all $h \in \mathbb{N}$, $\mathbf{f} \in \mathbb{R}^h$,*

$$\|\mathbf{f}\|_1 \leq \sqrt{h}\|\mathbf{f}\|_2.$$

*Proof.* Define the vectors:

$$\mathbf{u} = (|\mathbf{f}_1|, |\mathbf{f}_2|, \ldots, |\mathbf{f}_h|), \quad \mathbf{v} = (1, 1, \ldots, 1) \in \mathbb{R}^h.$$

Then,

$$\mathbf{u} \cdot \mathbf{v} = \sum_{i=1}^{h} |\mathbf{f}_i| = \|\mathbf{f}\|_1.$$

Applying the Cauchy-Schwarz inequality,

$$\|\mathbf{f}\|_1 = \mathbf{u} \cdot \mathbf{v} \leq \|\mathbf{u}\|_2 \|\mathbf{v}\|_2 \leq \sqrt{h}\|\mathbf{f}\|_2.$$

$\square$

## E Corollary of Theorem 2

**Corollary 1.** *If $\mathbf{f} \in \{0, 1\}^h$, then $\|\mathbf{z}(\mathbf{x})\|_2^2$ approximates the number of activated features with the same error bound of the theorem.*

*Proof of Corollary.* If $\mathbf{f} \in \{0, 1\}^h$, $\|\mathbf{f}\|_2^2 = \sum_{j=1}^{h} f_j^2 = \sum_{f_j \neq 0; j \in [h]} \{1\} = |\{j \mid f_j > 0, j \in [h]\}|$.

$\square$

---

[6]**Causal Linear Representation Hypothesis:** Another interpretation of the linear representation hypothesis is suggested by Park et al. (2023), demonstrating that *causal inner product* can unify the representation in both embedding and unembedding, which satisfies Riesz isomorphism even through non-linear softmax unembedding layer. However, interpreting LRH to incorporate the causality diverges from widely investigated context of SAE which does not involve softmax-invariance and thus Riesz isomorphism is trivially achieved by one-layer weights vectors of autoencoder network.

## F  Derivation of Upper Bound based on the JL Lemma

We consider a high-dimensional space where $h > d$. Let $\{\mathbf{b}_i\}_{i=1}^{h} \subset \mathbb{R}^h$ be an orthonormal basis, satisfying $\|\mathbf{b}_i\|_2 = 1$, and $\mathbf{b}_i \perp \mathbf{b}_j$ for $i \neq j$, where $i, j \in [h]$. By the JL lemma 1, there exists a linear mapping $\Psi(\cdot)$ such that $2(1 - \delta) \leq \|\Psi(\mathbf{b}_i) - \Psi(\mathbf{b}_j)\|_2^2 \leq 2(1 + \delta)$. Let $\Psi(\mathbf{b}_i) = \mathbf{v}_i$. Then, $\|\Psi(\mathbf{b}_i) - \Psi(\mathbf{b}_j)\|_2^2 = \|\mathbf{v}_i - \mathbf{v}_j\|_2^2 = \|\mathbf{v}_i\|_2^2 + \|\mathbf{v}_j\|_2^2 - 2\mathbf{v}_i \cdot \mathbf{v}_j$. The JL lemma is typically proven using $f(\mathbf{b}_i) = \frac{1}{\sqrt{d}} A \mathbf{b}_i$, where $A_{i,j} \sim \mathcal{N}(0, 1)$, as shown in Kakade (2010). Taking expectation, we obtain $\mathbb{E}[\|\mathbf{v}_i\|_2] = 1$. If we assume $\|\mathbf{v}_i\|_2 = 1$ as a baseline, then

$$2 - 2\delta \leq 2 - 2\varepsilon \leq 2 + 2\delta \quad \iff \quad |\varepsilon| \leq \delta,$$

where $\varepsilon$ denotes the quasi-orthogonality of $h$ vectors $\mathbf{v}_i$ in $d$-dimensional space by Definition 1. Based on the JL Lemma constant (Theorem 1), we can define:

$$\varepsilon_{\mathrm{JL}} := \delta = \sqrt{\frac{20 \ln h}{d}}.$$

## G  Unreliable Quasi-Orthogonalities of Pre-trained SAE Decoders

Since the JL lemma guarantees the existence of a linear transformation within a certain error range, the quasi-orthogonality of the ground-truth dictionary should not exceed this bound for a given hidden embedding. Our formalization based on the JL lemma answer the following question; why not simply use the quasi-orthogonality of the pre-trained decoder? While this seems reasonable, empirical evidence suggests otherwise. As in Leask et al. (2025), the maximum pairwise cosine similarity of SAE decoder weights tends to be very high, and thus exceeds the loose upper bound $\varepsilon_{\mathrm{JL}}$ which can be easily calculated (Appendix F). This indicates that SAEs are unable to learn decoders that achieve quasi-orthogonal decomposition to the extent permitted by the available dimensionality, suggesting the pre-trained decoders can be highly unreliable.

## H  Why $\varepsilon$LBO Cannot Be a Loss Function

Ignoring the normalization effect from the denominator, the numerator of $\varepsilon$LBO takes the form $\|\mathbf{z}\|_2^2 - \|\mathbf{f}\|_2^2$, which can be factorized as $(\|\mathbf{z}\|_2 - \|\mathbf{f}\|_2)(\|\mathbf{z}\|_2 + \|\mathbf{f}\|_2)$. Minimizing the second term $(\|\mathbf{z}\|_2 + \|\mathbf{f}\|_2)$ in this factorization is not only meaningless to reduce, but also introduces vulnerability to shrinkage effects (Rajamanoharan et al., 2024).

## I  Detailed Experimental Results

We performed all experiments on a single NVIDIA A100 GPU. Following Bussmann et al. (2024), each training iteration used 4,096 tokens from OpenWebText. We trained the models for 20k iterations (81,920,960 tokens in total). Table 1, 2, and 3 were measured using the last 100 batches.

## J  Evaluating the Designed Sparse Autoencoder: Top-AFA with $\varepsilon$LBO

In Figure 4, one may ask whether the multi-modal or long-tailed shape of the $\varepsilon$LBO distribution indicates not a more granular evaluation, but rather a noisier metric. While it is true that $\varepsilon$LBO is a lower bound theoretically susceptible to noise induced by superposition interference, such questions can be partially addressed by evaluating the proposed SAE using our proposed metric. Figure 6 highlights two observations: (1) $\varepsilon$LBO can be better minimized to lower values with Top-AFA, and (2) existing activation functions exhibit significantly poorer $\varepsilon$LBO distributions, not simply attributable to noise. Therefore, $\varepsilon$LBO reveals the presence of a theoretically grounded objective that current SAE training methods have neglected.

| Layer | Activation | $k$ | $\lambda_{\text{AFA}}$ | Sparsity (L0) | Normalized MSE $\pm\sigma$ |
|---|---|---|---|---|---|
| 6 | Top-AFA | – | 1/128 | 331.62 | $0.127769 \pm 0.002493$ |
| 6 | Top-AFA | – | 1/64 | 1118.46 | $0.034370 \pm 0.003631$ |
| 6 | Top-AFA | – | 1/32 | 2037.74 | $0.000179 \pm 0.000044$ |
| 6 | Top-AFA | – | 1/24 | 2133.73 | $0.000179 \pm 0.000047$ |
| 6 | Top-AFA | – | 1/16 | 2344.23 | $0.000176 \pm 0.000041$ |
| 6 | Top-AFA | – | 1/8 | 2862.85 | $0.000231 \pm 0.000098$ |
| 6 | Batch Top-k | 32 | – | 32.00 | $0.061787 \pm 0.000583$ |
| 6 | Batch Top-k | 64 | – | 64.00 | $0.048710 \pm 0.000464$ |
| 6 | Batch Top-k | 128 | – | 128.00 | $0.036358 \pm 0.000348$ |
| 6 | Batch Top-k | 256 | – | 256.00 | $0.023379 \pm 0.000231$ |
| 6 | Batch Top-k | 512 | – | 512.00 | $0.008044 \pm 0.000121$ |
| 6 | Batch Top-k | 1024 | – | 1024.00 | $0.000202 \pm 0.000021$ |
| 6 | Batch Top-k | 2048 | – | 2048.00 | $0.000207 \pm 0.000019$ |
| 6 | Batch Top-k | 4096 | – | 2968.77 | $0.000221 \pm 0.000031$ |
| 6 | Batch Top-k | 8192 | – | 2949.76 | $0.000224 \pm 0.000019$ |
| 6 | Top-k | 32 | – | 31.98 | $0.063715 \pm 0.000661$ |
| 6 | Top-k | 64 | – | 63.94 | $0.050356 \pm 0.000508$ |
| 6 | Top-k | 128 | – | 127.84 | $0.038570 \pm 0.000388$ |
| 6 | Top-k | 256 | – | 255.99 | $0.026100 \pm 0.000253$ |
| 6 | Top-k | 512 | – | 512.00 | $0.008793 \pm 0.000075$ |
| 6 | Top-k | 1024 | – | 1023.46 | $0.000199 \pm 0.000020$ |
| 6 | Top-k | 2048 | – | 2044.35 | $0.000207 \pm 0.000018$ |
| 6 | Top-k | 4096 | – | 2970.86 | $0.000219 \pm 0.000021$ |
| 6 | Top-k | 8192 | – | 2949.76 | $0.000224 \pm 0.000019$ |

Table 1: Detailed results for layer 6 in Figure 5.

| Layer | Activation | $k$ | $\lambda_{\text{AFA}}$ | Sparsity (L0) | Normalized MSE $\pm\sigma$ |
|---|---|---|---|---|---|
| 7 | Top-AFA | – | 1/128 | 1533.98 | $0.004476 \pm 0.000346$ |
| 7 | Top-AFA | – | 1/64 | 1930.60 | $0.000221 \pm 0.000028$ |
| 7 | Top-AFA | – | 1/32 | 2066.28 | $0.000199 \pm 0.000042$ |
| 7 | Top-AFA | – | 1/24 | 2180.70 | $0.000197 \pm 0.000057$ |
| 7 | Top-AFA | – | 1/16 | 2382.96 | $0.000193 \pm 0.000049$ |
| 7 | Top-AFA | – | 1/8 | 3016.48 | $0.000239 \pm 0.000082$ |
| 7 | Batch Top-k | 32 | – | 32.00 | $0.071729 \pm 0.000614$ |
| 7 | Batch Top-k | 64 | – | 64.00 | $0.056417 \pm 0.000491$ |
| 7 | Batch Top-k | 128 | – | 128.00 | $0.042214 \pm 0.000361$ |
| 7 | Batch Top-k | 256 | – | 256.00 | $0.026796 \pm 0.000234$ |
| 7 | Batch Top-k | 512 | – | 512.00 | $0.009608 \pm 0.000134$ |
| 7 | Batch Top-k | 1024 | – | 1024.00 | $0.000215 \pm 0.000016$ |
| 7 | Batch Top-k | 2048 | – | 2048.00 | $0.000222 \pm 0.000015$ |
| 7 | Batch Top-k | 4096 | – | 2974.06 | $0.000236 \pm 0.000022$ |
| 7 | Batch Top-k | 8192 | – | 2956.95 | $0.000239 \pm 0.000020$ |
| 7 | Top-k | 32 | – | 32.00 | $0.073273 \pm 0.000660$ |
| 7 | Top-k | 64 | – | 63.88 | $0.057877 \pm 0.000517$ |
| 7 | Top-k | 128 | – | 127.81 | $0.044168 \pm 0.000393$ |
| 7 | Top-k | 256 | – | 255.79 | $0.029050 \pm 0.000246$ |
| 7 | Top-k | 512 | – | 512.00 | $0.009737 \pm 0.000073$ |
| 7 | Top-k | 1024 | – | 1023.45 | $0.000214 \pm 0.000024$ |
| 7 | Top-k | 2048 | – | 2045.20 | $0.000221 \pm 0.000026$ |
| 7 | Top-k | 4096 | – | 2975.86 | $0.000239 \pm 0.000026$ |
| 7 | Top-k | 8192 | – | 2956.95 | $0.000239 \pm 0.000020$ |

Table 2: Detailed results for layer 7 in Figure 5.

| Layer | Activation | $k$ | $\lambda_{\text{AFA}}$ | Sparsity (L0) | Normalized MSE $\pm\sigma$ |
|---|---|---|---|---|---|
| 8 | Top-AFA | – | 1/128 | 1577.59 | $0.003457 \pm 0.000286$ |
| 8 | Top-AFA | – | 1/64 | 1973.62 | $0.000230 \pm 0.000026$ |
| 8 | Top-AFA | – | 1/32 | 2103.63 | $0.000203 \pm 0.000050$ |
| 8 | Top-AFA | – | 1/24 | 2195.08 | $0.000197 \pm 0.000047$ |
| 8 | Top-AFA | – | 1/16 | 2420.21 | $0.000193 \pm 0.000039$ |
| 8 | Top-AFA | – | 1/8 | 3019.70 | $0.000223 \pm 0.000042$ |
| 8 | Batch Top-k | 32 | – | 32.00 | $0.079982 \pm 0.000662$ |
| 8 | Batch Top-k | 64 | – | 64.00 | $0.063075 \pm 0.000524$ |
| 8 | Batch Top-k | 128 | – | 128.00 | $0.047696 \pm 0.000392$ |
| 8 | Batch Top-k | 256 | – | 256.00 | $0.029998 \pm 0.000254$ |
| 8 | Batch Top-k | 512 | – | 512.00 | $0.009727 \pm 0.000151$ |
| 8 | Batch Top-k | 1024 | – | 1024.00 | $0.000222 \pm 0.000044$ |
| 8 | Batch Top-k | 2048 | – | 2048.00 | $0.000236 \pm 0.000057$ |
| 8 | Batch Top-k | 4096 | – | 2950.05 | $0.000243 \pm 0.000023$ |
| 8 | Batch Top-k | 8192 | – | 2951.90 | $0.000247 \pm 0.000027$ |
| 8 | Top-k | 32 | – | 31.99 | $0.081479 \pm 0.000693$ |
| 8 | Top-k | 64 | – | 63.83 | $0.064612 \pm 0.000539$ |
| 8 | Top-k | 128 | – | 127.84 | $0.049319 \pm 0.000415$ |
| 8 | Top-k | 256 | – | 256.00 | $0.032053 \pm 0.000268$ |
| 8 | Top-k | 512 | – | 512.00 | $0.009610 \pm 0.000073$ |
| 8 | Top-k | 1024 | – | 1023.30 | $0.000220 \pm 0.000027$ |
| 8 | Top-k | 2048 | – | 2044.45 | $0.000228 \pm 0.000025$ |
| 8 | Top-k | 4096 | – | 2960.06 | $0.000247 \pm 0.000029$ |
| 8 | Top-k | 8192 | – | 2951.90 | $0.000247 \pm 0.000027$ |

Table 3: Detailed results for layer 8 in Figure 5.

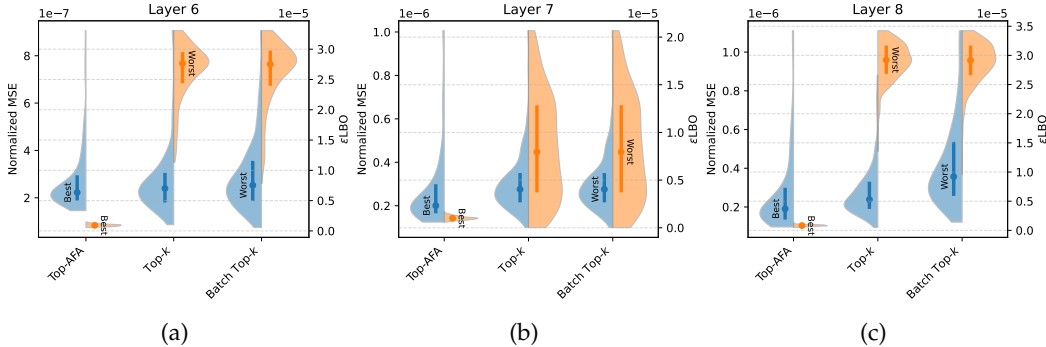

(a)                    (b)                    (c)

Figure 6: **Violin Plots for Comparing Different Activation Functions** (Lower values are better). Distributions are computed over 300 input sequences of length 128 from the OpenWebText dataset. The region of each violin captures 99% of the distribution to mitigate the influence of extreme outliers. The hyperparameter used for each layer corresponds to those that achieved the best reconstruction (the lowest NMSE) on the last batch of 20k training iterations. Interestingly, the $\varepsilon$LBO of Top-AFA can converge to a low value through training, while other activation functions, Top-$k$ and Batch Top-$k$, fail to achieve a low $\varepsilon$LBO.

## K   Complexity Analysis

Table 4 shows that Top-AFA achieves similar or better NMSE compared to other methods with training time that remains in a comparable range. To support our claim and explain the empirical results, we also conducted the complexity analysis. The time complexity of the encoding and decoding stages is summarized in Table 5. The key difference lies in the top-k selection mechanism, where sorting dominates, as detailed in Table 6. Since $k < h$, the

| Layer | Activation | L0 | NMSE | Time (min) |
|:---:|:---:|:---:|:---:|:---:|
| 6 | Top-AFA | 2037.29 | 0.000164 | 69.40 |
| 6 | Batch Top-k | 1024 | 0.000165 | 136.63 |
| 6 | Top-k | 1023.49 | 0.000174 | 69.98 |
| 7 | Top-AFA | 1930.10 | 0.000195 | 139.75 |
| 7 | Batch Top-k | 2957.23 | 0.000181 | 138.88 |
| 7 | Top-k | 2957.23 | 0.000181 | 70.61 |
| 8 | Top-AFA | 2195.77 | 0.000166 | 142.04 |
| 8 | Batch Top-k | 1024 | 0.000187 | 74.45 |
| 8 | Top-k | 1023.32 | 0.000177 | 71.90 |

Table 4: Layer-wise training time and reconstruction performance (NMSE), including L0 with the lowest NMSE on the last batch of 20k iterations. These results demonstrate that Top-AFA achieves training times comparable to those of other methods.

| Stage | Operation | Time Complexity |
|:---|:---|:---|
| Encoding | $\mathbf{x} \cdot W_{\mathrm{enc}}$ | $\mathcal{O}(Bdh)$ |
| Decoding | $\mathbf{f} \cdot W_{\mathrm{dec}}$ | $\mathcal{O}(Bkd)$ |

Table 5: Time complexity for encoding and decoding stages.

leading term for Top-AFA is $\mathcal{O}(Bhd + Bh\log h)$, which is not higher than other activation methods.

The memory complexity for each method is compared in Table 7, showing that all methods exhibit comparable complexity.

| Method | Top-k Selection Method | Time Complexity |
|---|---|---|
| Top-k | Sort per input ($B$ samples) | $\mathcal{O}(Bh \log h)$ |
| Batch Top-k | Sort across batch ($Bh$ entries) | $\mathcal{O}(Bh \log(Bh))$ |
| Top-AFA | Sort + cumulative sum + argmin per input | $\mathcal{O}(Bh \log h)$ |

Table 6: Time complexity of top-k selection mechanisms.

| Variable | Shape | Memory | (Batch) Top-k | Top-AFA |
|---|---|---|---|---|
| Input $\mathbf{x}$ | $B \times d$ | $\mathcal{O}(Bd)$ | ✓ | ✓ |
| Hidden activations $\mathbf{f}$ | $B \times h$ | $\mathcal{O}(Bh)$ | ✓ | ✓ |
| Reconstructed output $\hat{\mathbf{z}}$ | $B \times d$ | $\mathcal{O}(Bd)$ | ✓ | ✓ |
| Encoder weights $W_{\mathrm{enc}}$ | $d \times h$ | $\mathcal{O}(dh)$ | ✓ | ✓ |
| Decoder weights $W_{\mathrm{dec}}$ | $h \times d$ | $\mathcal{O}(dh)$ | ✓ | ✓ |
| Top-k indices / mask | $\leq B \times h$ | $\mathcal{O}(Bh)$ | ✓ | ✓ |
| Sort buffer | $B \times h$ | $\mathcal{O}(Bh)$ | — | ✓ |

Table 7: Memory complexity comparison across methods.

