# OpenReview forum: "Evaluating and Designing Sparse Autoencoders by Approximating Quasi-Orthogonality"
_colmweb.org/COLM/2025/Conference — COLM 2025_

### Official Review · Reviewer_8rCC · 2025-05-02

**Rating:** 6
**Confidence:** 4
**Ethics Flag:** 1

**Summary:**

The authors use the Liner Representation Hypothesis and the Superposition Hypothesis to make theoretical predictions related to the ortogonality of SAE features as well as predictions about the relation between the norms of the inputs and the norms of the activations.
With these theoretical insights they propose a new loss term and a new activation function, which can get competitive reconstructions when $L_0>10^3$ .

**Questions To Authors:**

Why should we care about the relation between f and z? The point of sparse autoencoders seems to be that that the reconstrution is good. Why should the norms have to match?

The claim is that $\epsilon LBO$ is more informative than just evaluating the MSE because the distribution has a more complex "shape". Could it just mean that the metric is more noisy, not informative or that is is evidence that the LRH and SH don't hold that strongly?

Sparsity greater than $10^3$ is probably useless and the learned features are probably not interpretable or at least as interpretable as neurons. Why use top AFA at all?

**Reasons To Accept:**

The authors create a nice framing of sparse autoencoders using the Linear Representation Hypothesis and the Superposition Hypothesis. The tools they take from high dimensional analysis allow them to define error bounds on the reconstruction of the sparse autoencoder.

**Reasons To Reject:**

The proposed architecture seems to fail to be relevant on the exact property that makes SAE interesting: find sparse interpretable features. The only reagion where top-AFA sparse autoencoders are competitive is at very high $L_0$, places where the learned features are probably as interpretable as neurons.
The claim that k is arbitrarily chosen is valid, not being obvious how to define the best K à priori, but this hyperparameter is substituted with a new one, $\lambda_{AFA}$, which the training process is very sensitive to as claimed by the authors.

---

> ### Author Response · Authors · 2025-06-03
>
> > Why should we care about the relation between f and z? The point of sparse autoencoders seems to be that the reconstruction is good. Why should the norms have to match?
>
> We thank you for this insightful question. One of the findings of our work is that, if the reconstruction is *perfect* ($D\mathbf{f} = \mathbf{z}$) in a widely-used single-layer SAE, the norms of z and the norms of true f must theoretically be approximately equal. Thus, if the reconstruction is simply *good*, checking whether the norms match can serve as a valuable diagnostic tool, especially since state-of-the-art methods like k-sparse autoencoders impose a strong and naïve inductive bias that exactly $k$ features should exist in every input. That is, when $||\mathbf{f}|| ≪ ||\mathbf{z}||$, we can now tell the SAE is under-activating the feature vector, implying that additional, nearly orthogonal features (as guaranteed to exist by the Johnson-Lindenstrauss lemma in high dimensions) should be added, not removed, to improve the reconstruction. When $||\mathbf{f}|| ≫ ||\mathbf{z}||$, it indicates either over-activation or poor orthogonality in the learned dictionary (For more details, see geometrical intuition in Appendix D).
>
> > The claim is that εLBO is more informative than just evaluating the MSE because the distribution has a more complex "shape". Could it just mean that the metric is more noisy, not informative or that it is evidence that the LRH and SH don't hold that strongly?
>
> We agree that having a complex distribution shape does not by itself make a metric better, and we will revise or delete any parts of the paper that might have suggested this. However, the concern that εLBO may simply be a noisier metric can be addressed by the additional experiments we conducted as shown in the table below. These are the results of evaluating our final experiments (Figure 4 and Table 1, 2, and 3) not only with MSE but also with our proposed εLBO metric:
>
> | Layer | Activation     | L0       | NMSE       | εLBO ± $\sigma (\times 10^{-6})$        |
> |-------|----------------|----------|------------|--------------------------|
> | 6     | Top-AFA        | 2037.29  | 0.000164   | 0.9274 ± 0.1815          |
> | 6     | Batch Top-k    | 1024     | 0.000165   | 26.2765 ± 5.0585         |
> | 6     | Top-k          | 1023.49  | 0.000174   | 26.4806 ± 4.4506         |
> | 7     | Top-AFA        | 1930.10  | 0.000195   | 1.0198 ± 0.1943          |
> | 7     | Batch Top-k    | 2957.23  | 0.000181   | 8.5929 ± 5.6435          |
> | 7     | Top-k          | 2957.23  | 0.000181   | 8.5929 ± 5.6435          |
> | 8     | Top-AFA        | 2195.77  | 0.000166   | 0.8210 ± 0.1367          |
> | 8     | Batch Top-k    | 1024     | 0.000187   | 28.8057 ± 3.5836         |
> | 8     | Top-k          | 1023.32  | 0.000177   | 29.0227 ± 3.1700         |
>
> Notably, top-AFA SAEs reliably attain substantially lower εLBO values compared to top-k variants, suggesting that εLBO can be effectively minimized without compromising reconstruction quality. This shows that low-noise εLBO is not impossible to achieve per se, but rather that conventional k-sparse autoencoders fail to learn εLBO effectively.
>
> > Sparsity greater than 10^3 is probably useless and the learned features are probably not interpretable or at least as interpretable as neurons. Why use top AFA at all?
>
> We agree that the usefulness of a sparsity as high as 1k-2k in a dictionary size of 12k is an important and open question. Although some studies limited $k$ to very small values less than 64 [1], recent work has also suggested that “sparsity is not a good proxy for interpretability” [2]. For example, [3] argues that maximizing sparsity is less important than minimizing description length, finding cases where sparser representations are less than optimal; thus, we do not feel that simply looking for small L0 is a fair comparison. Further, considering the diverse linguistic and semantic content that may be included in each of 128 tokens in each input context, it could actually be reasonable for the number of activated features to approach 2k-3k.
>
> [1] Bussmann, Bart, Patrick Leask, and Neel Nanda. "Batch Top-k sparse autoencoders." arXiv preprint arXiv:2412.06410 (2024).
>
> [2] Sharkey, Lee, et al. "Open Problems in Mechanistic Interpretability." arXiv preprint arXiv:2501.16496 (2025).
>
> [3] Ayonrinde, Kola, Michael T. Pearce, and Lee Sharkey. "Interpretability as Compression: Reconsidering SAE Explanations of Neural Activations." NeurIPS 2024 Workshop on Scientific Methods for Understanding Deep Learning.

---

> > ### Comment · Reviewer_8rCC · 2025-06-04
> >
> > > Notably, top-AFA SAEs reliably attain substantially lower εLBO values compared to top-k variants, suggesting that εLBO can be effectively minimized without compromising reconstruction quality. This shows that low-noise εLBO is not impossible to achieve per se, but rather that conventional k-sparse autoencoders fail to learn εLBO effectively
> >
> > I understand that top-AFA can better minimize εLBO that other models. What I don't think was shown in this article is that εLBO is measuring any of the things we end up caring about on downstream tasks, e.g, does the reconstructed output actually make the model behave as it should.
> >
> > >Although some studies limited to very small values less than 64 [1], recent work has also suggested that “sparsity is not a good proxy for interpretability” [2]. For example, [3] argues that maximizing sparsity is less important than minimizing description length, finding cases where sparser representations are less than optimal.
> >
> > It's good that the authors have brought up these articles, because I believe they don't argue for the point the authors are arguing for. [2] claims that we be careful when optimizing for sparsity because when the pressure is too high we start observing feature splitting, absorption, etc start happening. Still these are observed at L0s that are almost 2 orders of magnitude lower than the ones reported. [3], which [2] also cites, shows that the optimal MDL for the SAE they evaluate corresponds to an L0 of 10. Surelly the reviewers would agree then that an l0 of 10^3 will probably not be minimizing MDL. If the reviewers are able to show that they beat TopK SAEs at an MDL that is reasonable I will change my score to an accept, otherwise I think this work would have been an acceptable on if the results were shown as negative and not sold as beating baselines.
> >
> > An L0 of 2000 means the autoencoders have about the same L0 as the original residual stream. Is the point here that these "sparse" autoencoders are usefull not to make activations more sparse than the original ones, but oriented in better directions? Why would we train these instead of just using the residual stream?

---

> > > ### Author Response · Authors · 2025-06-07
> > >
> > > > What I don't think was shown in this article is that εLBO is measuring any of the things we end up caring about on downstream tasks
> > >
> > > We agree that evaluating the effectiveness of εLBO on downstream tasks is important, and should therefore be addressed next. However, such a study would go beyond the scope of the current paper, which focuses on revealing a theoretical link that $\ell_2$-norm of the sparse feature vector can be approximated with the $\ell_2$-norm of the dense vector and on empirically validating this link. Therefore, we believe this would warrant a separate paper.
> > >
> > > > [3], which [2] also cites, shows that the optimal MDL for the SAE they evaluate corresponds to an L0 of 10.
> > >
> > > There are two important distinctions regarding the L0 of 10 in [3]. First, the model studied in [3] was trained on MNIST, which has much lower complexity than real-world scale LLMs. Second, even considering Bloom (2024), a study cited in [3], such a low L0 can be simply achieved by heavily penalizing the L1 sparsity constraint term. This approach has an even weaker justification than our proposed coefficient $\lambda_{AFA}$ because simply increasing the coefficient will inevitably yield higher sparsity, damaging the reconstruction, without providing any desired range of the coefficient.
> > >
> > > > If the reviewers are able to show that they beat TopK SAEs at an MDL that is reasonable I will change my score to an accept
> > >
> > > We appreciate your suggestion of this interesting approach to justify our research. According to [3], MDL can be defined at a certain tolerance level, which in the case of SAEs can be measured in terms of reconstruction error. Thus, to compare TopK and Top-AFA SAEs, we compute MDL using the hyperparameter setting that achieves the tightest tolerance across different activation functions at each layer. The results show that top-AFA yields poorer MDL compared to top-k SAEs on layer 6 and 8, but achieves better MDL on layer 7.
> > >
> > >
> > > Layer 6:
> > > | Activation   | MDL (bits) |
> > > |--------------|-------------|
> > > | batchtopk    | 21079.0     |
> > > | **topk**     | **21068.6** |
> > > | topafa       | 41937.5     |
> > >
> > >
> > > Layer 7:
> > > | Activation   | MDL (bits) |
> > > |--------------|-------------|
> > > | batchtopk    | 60874.6     |
> > > | topk         | 60874.6     |
> > > | **topafa**   | **39731.1** |
> > >
> > >
> > > Layer 8:
> > > | Activation   | MDL (bits) |
> > > |--------------|-------------|
> > > | batchtopk    | 21079.0     |
> > > | **topk**     | **21065.1** |
> > > | topafa       | 45199.7     |
> > >
> > > Although our method does not always outperform the existing baseline, we believe the fact that top-AFA can outperform top-k SAEs in some cases, even though top-k SAEs are currently considered state-of-the-art, together with the stronger theoretical justification, makes our contribution worthy of publication to the research community.
> > > (Code is available at …/visualization/mdl.ipynb)
> > >
> > > > An L0 of 2000 means the autoencoders have about the same L0 as the original residual stream. Is the point here that these "sparse" autoencoders are usefull not to make activations more sparse than the original ones, but oriented in better directions? Why would we train these instead of just using the residual stream?
> > >
> > > Thank you for this sharp and fundamentally important question. Although this point was not explicitly discussed in the paper, it is a question we are also concerned about. If sparsity is viewed simply as the number of non-zero elements, we agree that there may appear to be little reason to train these SAEs. However, sparsity has also long been considered as the average activation probability of each feature, which reflects how frequently a neuron is active across the dataset [4]. Specifically, in a dense residual stream, each neuron is activated with virtually 100% probability. In contrast, for top-AFA, the activation probabilities across layer 6, 7, and 8 are 16.6%, 15.7%, 17.9%, respectively. This indicates a fundamental difference between the SAE feature vectors and the residual stream.
> > >
> > > [4] Ng, Andrew. "Sparse autoencoder." CS294A Lecture notes 72.2011 (2011): 1-19.

---

> > > > ### Comment · Reviewer_8rCC · 2025-06-09
> > > >
> > > > I remain concerned about the pratical usefullness of top-APA SAEs, but acknowledge that the authors have done significant work adressing my concerns, and have theoretical results worth mentioning to the community.
> > > >
> > > > I have raised my score.

---

> > > > > ### Author Response · Authors · 2025-06-10
> > > > >
> > > > > We thank you for your thoughtful engagement and feedback throughout the discussion.

---

> ### Author Response · Authors · 2025-06-03
>
> > The claim that k is arbitrarily chosen is valid, not being obvious how to define the best K à priori, but this hyperparameter is substituted with a new one, $\lambda$ , which the training process is very sensitive to as claimed by the authors.
>
> As we included in the Limitation section, we agree that we have introduced an alternative hyperparameter $\lambda_{AFA}$. Nevertheless, we believe our contribution still provides meaningful value to the research community, and we would like to clarify the distinction between λ and 𝑘 in more detail.
> First, $\lambda$ is a training-time regularization coefficient, whereas $k$ directly relates to inference-time because it refers to the number of active features for each input example. In this sense, $\lambda$ and $k$ start from fundamentally different ideas.
> Second, our approach allows the model to adaptively determine the effective number of active features for each input, rather than enforcing a fixed $k$ across all inputs. This reflects the intuition that different examples may require different amounts of feature activation, which a fixed $k$ cannot capture (and which is grounded mathematically by the theory in our paper).
> Therefore, we feel that replacing the need to arbitrarily set $k$ by the training-time hyperparameter $\lambda$ is an advantage of our approach that is motivated by the underlying ideas of sparse autoencoders.

---

> > ### Comment · Reviewer_8rCC · 2025-06-04
> >
> > > First, $\lambda$ is a training-time regularization coefficient, whereas $k$ directly relates to inference-time because it refers to the number of active features for each input example
> >
> > I agree with this, but think that this is actually worse that $\lambda$  is a training-time coefficient because it makes it harder to actually train a saprse autoencoder with the desired level of sparsity.
> >
> > > Second, our approach allows the model to adaptively determine the effective number of active features for each input, rather than enforcing a fixed $k$ across all inputs.
> >
> > So does BatchTopK, meaning that you traded an hyperparameter that easily controls for sparsify both at train and inference time for one that does not.

---

> > > ### Author Response · Authors · 2025-06-07
> > >
> > > We sincerely thank you for your valuable feedback. Below is what we have experimentally and conceptually verified in response to the points you raised.
> > >
> > > > this is actually worse that λ is a training-time coefficient because it makes it harder to actually train a saprse autoencoder with the desired level of sparsity.
> > >
> > > Our goal is not necessarily to make it *easier* to train a sparse autoencoder with a desired level of sparsity. Rather, our motivation stems from the observation that no one truly knows the ground-truth level of sparsity, and our method is designed to let the model determine this dynamically.
> > >
> > > > So does BatchTopK
> > >
> > > Even BatchTopK requires fixing the average k within a batch. Despite the fact that we do not know the ground-truth, BatchTopK still requires specifying the expected sparsity level before training, not obviating a core limitation that our method seeks to address.

---

### Official Review · Reviewer_Ji2v · 2025-05-12

**Rating:** 7
**Confidence:** 2
**Ethics Flag:** 1

**Summary:**

This paper introduces a method called approximate feature activation (AFA) that is shown theoretically to approximate the magnitude of the (unknown) ground-truth feature vector, under certain common assumptions (which might be practical, though they are probably too strong in reality). AFA gives rise to a new metric for SAE and a nice visualization; and a new flavor of SAE, which seems promising in a very simple experiment.

**Questions To Authors:**

Minor presentation comments:
- For a paper that is so crisp and economical with its notation, built from the ground up, it is surprisingly free with obscure jargon that is specific to this subfield e.g. "meaningfully aligned" (line 37) and "residual stream" (line 79).
- Unlike the rest of the paper, the abstract is a big mess; more of a story about how the paper is structured than what the paper delivered. Weirdly, the arbitrariness of $k$ is introduced in the first sentence, then completely ignored for the remainder of the abstract until almost the end, with nary a bit of logical connection in between. I thought this paper would be an impossible mess based on the abstract but it was (mostly) pretty clear. Please rewrite the abstract for clarity.

**Reasons To Accept:**

- The paper is well-written, given the theoretical subject matter. Although I'm not an expert in this area, I mostly understood what was going on, with a bit of effort.
- The paper provides new methodology and metrics that should help the interpretability community understand SAEs and how well they align with underlying features. It seems plausible to me that this analysis could help others with their ongoing research, (note that I'm an outsider to this field, so this is just an educated guess on my part, based on the context in the paper).

**Reasons To Reject:**

- I am a bit worried that this community and its ecosystem (nicely outlined in the background section) relies on models and datasets that have accumulated around increasingly out of date models. I don't think this is a major problem for a mostly theoretical paper like this, that is really exploring a new idea in the space. But it does leave open the natural question of what happens when one applies these ideas to a more current model. I appreciate that this is difficult to do for many researchers, and also that this particular paper needs to work with what is available.
- I also suspect that the strict hypotheses that the method relies on are too strong for real world data (e.g. that "ground truth" features, even if we could completely identify them, are "nearly orthogonal"). I think that's generally ok so long as it's acknowledged that mathematical niceties don't often hold up in their entirety in the real world. All models are wrong, some models are useful...

---

> ### Author Response · Authors · 2025-06-03
>
> We sincerely thank you for your positive evaluation.
>
> > For a paper that is so crisp and economical with its notation, built from the ground up, it is surprisingly free with obscure jargon that is specific to this subfield e.g. "meaningfully aligned" (line 37) and "residual stream" (line 79).
>
> We thank you for pointing out terminology that is not clear to readers. Regarding “meaningfully aligned,” we will revise the sentence to more clearly emphasize a limitation in prior work: although existing methods aim to construct sparse features corresponding to inputs, the selection of sparsity levels in these methods is independent of the inputs themselves.
>
> For the term “residual stream”, we will add a clarification in the main text to explicitly define it as the hidden embedding vector passed through the residual connections between transformer layers. We will also link this to Appendix B, where the technical background is described in more detail, and point to existing literature that uses this term to ensure consistency with prior work.
>
> > Unlike the rest of the paper, the abstract is a big mess; more of a story about how the paper is structured than what the paper delivered. Weirdly, the arbitrariness of $k$ is introduced in the first sentence, then completely ignored for the remainder of the abstract until almost the end, with nary a bit of logical connection in between. I thought this paper would be an impossible mess based on the abstract but it was (mostly) pretty clear. Please rewrite the abstract for clarity.
>
> We agree and thank you for this candid feedback. We have rewritten it as below.
>
>
> **Updated Abstract**
>
> Sparse autoencoders (SAEs) are widely used in mechanistic interpretability research for large language models; however, the state-of-the-art method of using $k$-sparse autoencoders lacks a theoretical grounding for selecting the hyperparameter $k$ that represents the number of nonzero activations, often denoted by $\ell_0$. In this paper, we reveal a theoretical link that $\ell_2$-norm of the sparse feature vector can be approximated with the $\ell_2$ norm of the dense vector with a closed form error, which allows sparse autoencoders to be trained without the need to manually determine $\ell_0$. Specifically, we validate two applications of our theoretical findings. First, we introduce a new methodology that can assess the feature activations of pre-trained SAEs by computing the theoretically expected value from the input embedding, which has been overlooked by existing SAE evaluation methods or loss functions. Second, we introduce a novel activation function, top-AFA, which builds upon our formulation of approximate feature activation (AFA). This function enables top-$k$ style activation without requiring a constant hyperparameter $k$ to be tuned, dynamically determining the number of activated features for each input. By training SAEs on three intermediate layers to reconstruct GPT2 hidden embeddings for over 40 million tokens from the OpenWebText dataset, we demonstrate the empirical merits of this approach and compare it with current state-of-the-art $k$-sparse autoencoders.

---

> > ### Comment · Reviewer_Ji2v · 2025-06-03
> >
> > Thanks for the clarification.

---

### Official Review · Reviewer_b4e1 · 2025-05-13

**Rating:** 7
**Confidence:** 2
**Ethics Flag:** 1

**Summary:**

The paper analyzes Sparse Autoencoders (SAEs) with top-k style activation functions theoretically and establishes new relationships between LLM hidden embeddings and SAE feature vectors. With this, the authors "measure" how well the SAE feature vectors are activated for a given input via ZF plots and with a new metric called Approximate Feature Activation (AFA).

**Reasons To Accept:**

- The paper makes a number of contributions to theoretical fundamentals of mechanistic interpretability via SAEs. While I am not very familiar with the literature, the findings looked correct to me.

- The proposed method, top-AFA SAE, eliminates the need to tune k in top-k style activations.

- Empirical results show that top-AFA SAE outperforms top-k and batch top-k SAEs.

**Reasons To Reject:**

I can't find a strong weakness but I am also not an expert in this topic.

---

> ### Author Response · Authors · 2025-06-02
>
> We sincerely appreciate your verification of our theoretical contributions and positive evaluation of our overall findings. Your summary suggests a strong grasp of the paper. Are there any clarifications we could provide that would help raise your confidence in the assessment? Thank you again for your time spent reviewing our paper.

---

### Official Review · Reviewer_QxGv · 2025-05-26

**Rating:** 4
**Confidence:** 4
**Ethics Flag:** 1

**Summary:**

The authors introduce Approximate Feature Activation (AFA) to evaluate and design sparse autoencoders (SAEs) by connecting input embeddings to feature activations through quasi-orthogonality constraints. Furthermore, they propose several evaluation metrics, such as  ZF plot for visualization and the ε_LBO metric for evaluation of SAEs and a top-AFA SAE architecture that adaptively selects sparsity levels without hyperparameter tuning.

The authors provide solid theoretical foundations with formal proofs and empirical validation for AFA, however, the dense mathematical exposition may hinder accessibility for a general audience. The connection between theoretical concepts and practical applications could be clearer. The adaptive sparsity selection approach is novel, though the core ideas build incrementally on existing work. While the theoretical insights are valuable for understanding SAE behavior, the practical impact appears limited due to modest performance gains and additional complexity in implementation, especially when recent work has shown limited gains SAEs provide as compared to simple baselines for practical tasks. The paper can also benefit from significant restructuring of the main content, such as describing the intuitive interpretation of the top-AFA function and the ZF plot, along with Top-AFA algorithm in the main article and discussions of the strengths and limitations of the proposed architecture and the AFA framework to be of broader interest to the COLM  community.

**Questions To Authors:**

1. The reviewer’s main concern is with the introduced loss term. The authors claim to eliminate the need for tuning k, but introduce λ in the loss, which appears equally critical. This seems to lessen the strength of the main claim about reducing hyperparameter sensitivity. More clarity on the justification of the choice of the additional loss term would also be helpful.

2. The top-AFA algorithm involves sorting and cumulative sum operations for each input. How does this computational overhead compare to simple top-k selection? Is the improved performance worth the added complexity?

3. How statistically significant are the performance improvements? What are the confidence intervals, and how many random seeds were used? This might help provide clarity on the improvement claims.

4. Does the proposed approach improve the interpretability of learned features compared to standard top-k/batch top-k SAEs? Do the adaptively selected features provide better semantic coherence?

5. What are the training time and memory requirements compared to baseline methods?

6. How do the top-AFA SAEs perform on downstream tasks that measure feature quality, such as those in SAE Bench?

**Reasons To Accept:**

1. The authors provide a theoretical foundation as a connection between input embeddings and feature activations in SAEs and provide metrics that may be valuable for evaluating SAE performance beyond traditional reconstruction-based metrics.

2. The top-AFA approach provides theoretical justification for adaptive sparsity selection, addressing a known limitation of SAEs with top-k based activations.

3. The work includes a thorough empirical comparison with standard SAEs.

**Reasons To Reject:**

1. While the AFA based approach provides an alternative to standard top-k activation based SAEs, the performance improvements are marginal and inconsistent across layers, raising questions about practical significance, particularly since the method only outperforms baselines on specific layers (6 and 8) but not others, and for the specific choice of loss hyperparameter of 1/32

2. The work would benefit from some comparison of the top-AFA SAE features with top-k or batch top-k SAE features, and their effectiveness on downstream tasks such as monitoring or controlling safety related behaviors.

3. The addition of an additional AFA loss term might shift the optima of the loss objective from proper reconstruction, so it might be worth exploring the effect of the additional loss term. This might also point to why top-AFA might not outperform its top-K counterparts except for specific hyperparameter settings.

---

> ### Author Response · Authors · 2025-06-03
>
> We thank the reviewer for the detailed and constructive feedback.
>
> **Restructuring of the Main Content**
>
> We agree that the presentation can be improved. In the revised version, we will move the description of the top-AFA algorithm (currently Algorithm 1) and its geometric intuition (from Appendix D) into the main body, immediately after Theorem 2. We believe this will better connect the theoretical framework with its practical implementation and improve the paper's overall readability.
>
> **Induced Loss Term**
>
> Question 1
> > The reviewer’s main concern is with the introduced loss term. The authors claim to eliminate the need for tuning k, but introduce λ in the loss, which appears equally critical. This seems to lessen the strength of the main claim about reducing hyperparameter sensitivity. More clarity on the justification of the choice of the additional loss term would also be helpful.
>
> > The addition of an additional AFA loss term might shift the optima of the loss objective from proper reconstruction, so it might be worth exploring the effect of the additional loss term. This might also point to why top-AFA might not outperform its top-K counterparts except for specific hyperparameter settings.
>
> We would first like to clarify that the added loss term does not shift the optimum of the reconstruction objective. The L2 reconstruction loss is minimized when $D\mathbf{f}=\mathbf{z}$, and under this condition, Theorem 2 guarantees that $\mathbb{E}_\varepsilon[\|f\|] = \|z\|$. We will add this explanation to the revised paper to better clarify the role and effect of the loss term.
> Further, as noted in our Limitations section, we acknowledge that the justification for introducing the additional loss term could be strengthened, and a deeper theoretical and empirical analysis of this loss is left for future work.

---

> ### Author Response · Authors · 2025-06-03
>
> Question 2 and 5
> > The top-AFA algorithm involves sorting and cumulative sum operations for each input. How does this computational overhead compare to simple top-k selection? Is the improved performance worth the added complexity?
> > What are the training time and memory requirements compared to baseline methods?
>
> We agree that computational complexity is an important consideration and took care to address it during development. The following table provides layer-wise training time and the best reconstruction performance (NMSE), demonstrating that Top-AFA is competitive in practice:
> | Layer | Activation   | L0       | NMSE       | Time (min)   |
> |-------|--------------|----------|------------|--------------|
> | 6     | Top-AFA      | 2037.29  | 0.000164   | 69.40        |
> | 6     | Batch Top-k  | 1024     | 0.000165   | 136.63       |
> | 6     | Top-k        | 1023.49  | 0.000174   | 69.98        |
> | 7     | Top-AFA      | 1930.10  | 0.000195   | 139.75       |
> | 7     | Batch Top-k  | 2957.23  | 0.000181   | 138.88       |
> | 7     | Top-k        | 2957.23  | 0.000181   | 70.61        |
> | 8     | Top-AFA      | 2195.77  | 0.000166   | 142.04       |
> | 8     | Batch Top-k  | 1024     | 0.000187   | 74.45        |
> | 8     | Top-k        | 1023.32  | 0.000177   | 71.90        |
>
> (The experiment was conducted using a single A100 GPU.)
>
> These results show that Top-AFA achieves similar or better NMSE compared to other methods with training time that remains in a comparable range. To support our claim and explain the empirical results, we also conducted the complexity analysis as follows.
>
> | Stage             | Operation                    | Time Complexity |
> |------------------|------------------------------|-----------------|
> | Encoding         | $\mathbf{x} \cdot W_{enc}$                     | O(Bdh)          |
> | Decoding         | $\mathbf{f} \cdot W_{dec}$            | O(Bkd)          |
>
> The key difference lies in the top-k selection mechanism, where sorting dominates:
>
> | Method         | Top-k Selection Method                 | Time Complexity        |
> |----------------|-------------------------------------------|------------------------|
> | Top-k          | Sort per input (B samples)                  | O(Bh log h)            |
> | Batch Top-k    | Sort across batch (Bh entries) | O(Bh log (Bh))         |
> | Top-AFA        | Sort + cumulative sum + argmin per input    | O(Bh log h)            |
>
> Since $k < h$, the leading term for Top-AFA is $O(Bhd + Bh \log h)$, which is not higher than other activation methods.
>
> Memory complexity can be analyzed as follows, and all methods exhibit comparable complexity, which is $O(Bd + dh + Bh)$:
>
> | Variable                      | Shape   | Memory     | Top-k | Batch Top-k | Top-AFA |
> |-------------------------------------|----------------------|------------|:-----:|:-----------:|:-------:|
> | Input x                             | B × d                | O(Bd)      |  ✓    |      ✓      |    ✓    |
> | Hidden activations (acts)           | B × h                | O(Bh)      |  ✓    |      ✓      |    ✓    |
> | Reconstructed output (acts @ W_dec) | B × d                | O(Bd)      |  ✓    |      ✓      |    ✓    |
> | Encoder weights (W_enc)             | d × h                | O(dh)      |  ✓    |      ✓      |    ✓    |
> | Decoder weights (W_dec)             | h × d                | O(dh)      |  ✓    |      ✓      |    ✓    |
> | Top-k indices / mask                | ≤ B × h          | O(Bh)      |  ✓    |      ✓      |    ✓    |
> | Sort buffer (argsort and cumsum) | B × h                | O(Bh)      |   –   |      –      |    ✓    |
>
> Thank you again for raising this important point. We will include this analysis in the appendix of the revised version.

---

> ### Author Response · Authors · 2025-06-03
>
> Question 3
> > How statistically significant are the performance improvements? What are the confidence intervals, and how many random seeds were used? This might help provide clarity on the improvement claims.
>
> This is an excellent and important question. In line with prior SAE training setups that use online learning [1, 2], our evaluation was based on the L2 reconstruction loss measured on the final mini batch after 2K training steps. Typically, random seeds are not varied in this setting because SAEs are simple single-layer models with a large number of randomly initialized weights, and the effect of random seeds tends to be negligible. Nevertheless, we agree that assessing statistical significance is valuable. While our final mini batch included 4,096 input examples, we did not perform statistical testing when aggregating the results. To address this, we conducted a t-test over the final 100 batches, comparing top-AFA with top-k and batch top-k baselines:
>
> | Layer | Group 1 (Top-AFA) | Group 2          | Mean 1     | Var 1      | Mean 2     | Var 2      | t-stat   | p-value   |
> |-------|-------------------|------------------|------------|------------|------------|------------|----------|-----------|
> | 6     | Top-AFA           | Top-K            | 1.79e-04   | 1.96e-09   | 1.99e-04   | 4.14e-10   | -4.20    | 4.78e-05  |
> | 6     | Top-AFA           | Batch Top-K      | 1.79e-04   | 1.96e-09   | 2.02e-04   | 4.61e-10   | -4.67    | 6.89e-06  |
> | 7     | Top-AFA           | Top-K            | 2.21e-04   | 7.92e-10   | 2.39e-04   | 3.96e-10   | -5.19    | 5.62e-07  |
> | 7     | Top-AFA           | Batch Top-K      | 2.21e-04   | 7.92e-10   | 2.39e-04   | 3.96e-10   | -5.19    | 5.62e-07  |
> | 8     | Top-AFA           | Top-K            | 1.97e-04   | 2.24e-09   | 2.28e-04   | 6.16e-10   | -5.77    | 4.44e-08  |
> | 8     | Top-AFA           | Batch Top-K      | 1.97e-04   | 2.24e-09   | 2.22e-04   | 1.93e-09   | -3.78    | 2.04e-04  |
>
> The results show that the MSE improvements from top-AFA are statistically significant. Notably, even for Layer 7, where performance initially appeared weaker, the extended evaluation across more batches indicates that top-AFA actually outperforms the alternatives when tested more thoroughly. We plan to include these experimental settings and results in the revised version of the paper to strengthen our main experimental claims. We sincerely thank the reviewer for raising this point, and it helped us more clearly demonstrate the empirical contribution of our method.
>
>
> Question 4 and 6:
> > Does the proposed approach improve the interpretability of learned features compared to standard top-k/batch top-k SAEs? Do the adaptively selected features provide better semantic coherence?
>
> > How do the top-AFA SAEs perform on downstream tasks that measure feature quality, such as those in SAE Bench?
>
> Thank you for the question. Evaluating performance on downstream tasks such as those in SAE Bench is not part of the original motivation or scope of our work, which focuses primarily on theoretical insights and architectural analysis. We agree that such evaluations would be valuable for understanding the practical utility of our method, and we leave this, as well as a qualitative analysis of learned features, as an important direction for future work.
>
> [1] https://en.wikipedia.org/wiki/Online_machine_learning
>
> [2] Bussmann, Bart, Patrick Leask, and Neel Nanda. "Batch Top-k sparse autoencoders." arXiv preprint arXiv:2412.06410 (2024).

---

> ### Author Response · Authors · 2025-06-07
>
> We thank the reviewer again for the valuable feedback. We would like to gently remind you that the experimental results we shared address your concerns and further support the contributions of our work. As we near the end of the discussion phase, we respectfully ask if you would consider revisiting your evaluation in light of these clarifications and additional results.

---

### Author Response · Authors · 2025-06-10

We would like to thank all reviewers for their time and consideration. Reviewers **b4e1** and **Ji2v** both gave **scores of 7 (good paper, accept)**, and their summaries demonstrate a clear and accurate understanding of our contributions. Reviewer **8rCC** engaged with the paper with exceptional scrutiny, raising sharp and foundational questions regarding the usefulness of the proposed metric εLBO, the appropriate level of sparsity, and the validity of our claims through the lens of information theory, such as minimum description length (MDL). In the additional experiments measuring MDL, top-AFA outperformed top-k baselines in layer 7, and the reviewer concluded that this paper includes results **worth mentioning to the community**. Lastly, in response to Reviewer **QxGv**’s concern about statistical significance,  we conducted additional hypothesis testing, which further strengthened our claims. Notably, we found that **our method, top-AFA, consistently achieves significantly better reconstruction across all tested layers**. Once again, we sincerely appreciate the reviewers’ engagement and thoughtful input during the discussion.

---

### Decision · Program_Chairs · 2025-07-08

**Decision:**

Accept

**Comment:**

This paper proposes AFA, a method to approximate the magnitude of the ground truth sparse feature vector. They use this proposed method for both designing top-k SAEs with adaptively selected k and for evaluating SAE reconstruction performance. The contributions are a mix of theory, describing bounds on reconstruction, and empirical/methods work on applying AFA.

Pros:
- Nice formal treatment of SAEs in the context of the assumptions behind them like the LRH. Avoids pitfalls around informal treatment of the LRH by using a precise definition in line with the prevalent meaning.
- Potentially useful metrics introduced, good empirical confirmation.
- Theoretically justifies an adaptive hyperparameter selection method for SAEs.

Cons:
- Some assumptions may be too strong (near orthogonality) and the setting is small models, but both are common in SAE work.
- Effects may not be practically substantial (though they are confirmed to be statistically significant)